# Descriptive regression tree analysis of intersecting predictors of adult self-rated health: Does gender matter? A cross-sectional study of Canadian adults

**Afshin Vafaei**[1,2]*, **Jocelyn M. Stewart**[3], **Susan P. Phillips**[2,3]

**1** School of Health Studies, Western University, London, ON, Canada, **2** Department of Public Health Sciences, Queen's University, Kingston, ON, Canada, **3** Department of Family Medicine, Queen's University, Kingston, ON, Canada

* avafaei2@uwo.ca

## Abstract

### Background

While self-rated health (SRH) is a well-validated indicator, its alignment with objective health is inconsistent, particularly among women and older adults. This may reflect group-based differences in characteristics considered when rating health. Using a combination of SRH and satisfaction with health (SH) could capture lived realities for all, thus enabling a more accurate search for predictors of subjective health. With the combined measure of SRH and SH as the outcome we explore a range of characteristics that predict high SRH/SH compared with predictors of a low rating for either SRH or SH.

### Methods

Data were from the Canadian General Social Survey 2016 which includes participants 15 years of age and older. We performed classification and regression tree (CRT) analyses to identify the best combination of socioeconomic, behavioural, and mental health predictors of good SRH and health satisfaction.

### Results

Almost 85% of the population rated their health as good; however, 19% of those had low SH. Conversely, about 20% of those reporting poor SRH were, none-the-less, satisfied. CRT identified healthy eating, absence of a psychological disability, no work disability from long-term illness, and high resilience as the main predictors of good SRH/SH. Living with a spouse or children, higher social class and healthy behaviours also aligned with high scores in both self-perceived health measures. Sex was not a predictor.

**Data Availability Statement:** GSS data and analytical articles published by Statistics Canada are available to all interested parties: https://

www150.statcan.gc.ca/n1/pub/89f0115x/
89f0115x2013001-eng.htm#a5.

**Funding:** AV was supported by The Canadian Institutes of Health Research, award number 161787. The funders had no role in study design, data collection and analysis, decision to publish, or preparation of the manuscript.

**Competing interests:** The authors have declared that no competing interests exist.

**Abbreviations:** SRH, Self-rated Health; SH, Satisfaction with health; CRT, Classification and regression tree; SES, Socio-economic status; GSS, General Social Survey.

## Conclusions

Combining SRH and SH eliminated sex as a predictor of subjective health, and identified characteristics, particularly resilience, that align with high health and well-being and that are malleable.

## Background

Adults, and particularly those who are older, often rate their health as very good despite multimorbidity [1, 2], demonstrating their perception that the determinants of health and well-being extend beyond diagnosis and treatment of illness [3, 4]. If health is more than the absence of disease [5] what might maximize this state of well-being? We will explore the attributes of Canadians whose subjective rating of their health (SRH) and satisfaction with it are high, to identify characteristics of this group and, particularly, any that could be fostered via social policies or programs.

A strength of the indicator, SRH, is its potential to capture more than objective counts of diagnoses. Although such counts do predict longevity they give a limited picture of day-to-day health and well-being. Such a picture more clearly depicts how life is lived rather than only when it might end. SRH, at times, does present a composite picture of disease burden, mental health status, mortality, and the impact of social circumstances [3–9]. The subjective nature of the measure can allow for individualized responses that incorporate aspects of well-being and context [4]. However, that subjectivity also inserts an elasticity into characteristics considered in rating of health, producing inconsistent interpretations across individuals and groups. For example, SRH is often more predictive of mortality in men than women [10, 11]. Perhaps the experiences women consider when rating their health embrace social and contextual factors while men focus more specifically on number and nature of diagnoses [12, 13] The alignment of SRH and longevity or number of serious diseases also tends to vary with age. Older adults report relatively high SRH despite morbidity and looming mortality [1, 10]. These inconsistencies suggest complex variations in how perceptions of health, counts of diseases and lived circumstances differ among groups or individuals. Such inconsistencies can have a very real impact on subjective ratings of health.

What, then, might be a more universal measure of current health and well-being, one that is subject to less variability in interpretation of the question? There is evidence that satisfaction with life and health measures something different from SRH [14] but also aligns with current and subsequent health outcomes [15] and mortality [16]. Of importance from a policy perspective, life satisfaction is a *health asset* [17] that can be modified to improve physical health and well-being [18, 19].

Adding responses about satisfaction with health (SH) to SRH may insert aspects of lived realities overlooked by some when rating their health, and thus correct for inconsistencies noted earlier (e.g. by age and sex/gender) in how subjective health is reported. We use the term, sex/gender here as there is almost certainly an interaction between biologic sex and the lived social realities associated with being a man or a woman that we will refer to as gender [20, 21]. Combinations of SRH and SH might provide a broader and deeper picture of characteristics of those with highest reported current health and well-being, characteristics that augment health assets and that can be fostered.

There is extensive evidence that economic deprivation is among the strongest risks to health [22, 23], although less so for women than men [24] Equalizing socio-economic status (SES) is, however, neither politically nor practically straightforward [25]. Of particular interest to

policy-makers are health promoting circumstances or qualities that are malleable. For example, there is a nascent literature demonstrating that individual resilience is tied to satisfaction with life and health, can promote health, well-being and longevity [26–29] and can be fostered throughout life. Multiple definitions of resilience exist. In this context we define it as the process of positively adapting to adversity, trauma, threats, or significant distress [28].

Determining the characteristics of individuals reporting high SRH and high SH, particularly those that can be augmented via individual or social interventions without large political shifts, could guide policies and practices that extend beyond medical prevention and treatment. Despite a large body of evidence on predictors of SRH, to the best of our knowledge there is no study that directly identifies characteristics of individuals with high levels of both SRH and SH.

We aimed to identify medical and sociodemographic characteristics of Canadian adults with the highest SRH and SH, that is, those whose health and perceptions of well-being in the broadest sense, are greatest. This is novel and exploratory research, that by classifying a whole population into subgroups with the highest probability of the outcome, examines whether that subgroup's characteristics can be fostered in others. Our methodology was intentionally chosen to make no *a priori* assumptions, treat all factors equally, recognize the interconnections of characteristics, and let the data 'speak'.

We used data from the Canadian General Social Survey (GSS) to 1. examine the interplay between self-rated health and satisfaction with health, and 2. identify potential predictors of a high rate of both SRH and SH. The GSS is a well-established national survey that has been used to examine the impacts of caregiving [30], social capital [31], physical activity [32], and immigration status [33] on various aspects of health. To the best of our knowledge, no multi-dimensional models acknowledging the interconnectedness of SRH and SH have been described using GSS data. We performed Classification and Regression Tree (CRT) modelling to identify predictors of a high rating of this combined measure of SRH and SH. This exploratory methodology can identify interconnected (or intersecting) and non-linear relationships among social and medical predictors [34–36] by classifying the survey population into any combination of characteristics that predict the outcome of interest. As an example, it is possible to determine whether different combinations of SES, resilience and ability to work shape SRH/SH differently. This analytic approach enabled identification of specific behaviours, groups, or sub-groups for whom interventions could improve overall health and wellbeing.

## Methods

### Data source

The Canadian General Social Survey (GSS) 2016 (Cycle 30: Canadians at work and home) was used. This electronic or telephone-based survey is publicly available from Statistics Canada. Data were collected from August to December, 2016 and included Canadians aged 15 years and older, but excluded the Yukon, Northwest Territories and Nunavut, and residents of institutions. The sampling frame was created linking several sources, such as the Census of population, administrative data files and billing files. Sampling was completed by dividing provinces into geographic strata and identifying a representative number of participants. The response rate was 50.8%. We did not weight data given that this approach has the potential to over-amplify some populations while under-representing others, especially with methods such as CRT, which require higher accuracy in measurements [34–36]. As such, conclusions drawn represent those individuals sampled in the dataset and not the population of Canada as a whole. The analyzed data were deidentified by removing all personal and geographic identifiers.

## Outcome variables: SRH and satisfaction with health

SRH was determined by asking: "In general, would you say your health is. . ." Possible responses were 'excellent', 'very good', 'good', 'fair', 'poor' or 'don't know'. This variable was dichotomized by setting the cut-point between good and fair categories as per precedents [1, 3].

Level of satisfaction with health was assessed by a question derived from the Personal Well-Being Index and UK Office of National Statistics 2011 Opinion Survey and used a Likert scale from 1 to 10 (1 = not at all satisfied, 10 = completely satisfied). SH was also dichotomized by setting a cut-point of greater than or equal to 7 for 'good' satisfaction with health and less than or equal to 6 for 'poor' satisfaction with health. This cut-point was selected given the apparent bimodal distribution of responses and in order to provide efficiency in statistical analysis (see S1 File).

The combination of these variables was further categorized into two groups for CRT analysis defined as those with: 1. good SRH and high SH, and 2. a low rating of either indicator (i.e., low SRH-high HS, high SRH-low SH and low SRH-low SH). We propose that individuals with dissatisfaction with their health and/or poor SRH could benefit from interventions aimed at improving these factors and therefore the latter three categories described were grouped for the analysis.

## Explanatory variables

Age of respondents was categorised in 10-year groupings ('15–24','25–34', . . .) in the initial analysis. These data were then dichotomized to include respondents greater than or equal to age 65 or less than 65 years old. Sex was a dichotomous (male/female) and based on participant's self-identification.

Following Statistics Canada, areas with a core population of >50,000 inhabitants and with one or more neighbouring municipalities with a population of at least 100,000 were defined as 'urban'. All other regions were considered 'rural', with the exception of Canada's smallest province, PEI, which was recorded separately.

Perceived social class was measured by asking "People sometimes describe themselves as belonging to a particular social class. Which class would you describe yourself as belonging to?", adapted from the *World Values Survey*. Possible answers included: 'upper class', 'upper-middle class', 'middle class', 'lower-middle class', and 'lower class'. Education level was self-reported as: 'less than high school', 'high school diploma or trade certificate', 'college/CEGEP/other non-university diploma', 'university below bachelor's level', 'bachelor's degree', 'university certificate, diploma, degree above the BA level'. Annual family income was categorized as '<$25000', '$25000–49999', '$50000–74999', '$75000–99999', '$100000–124999' and '>$125000' (Canadian). Living arrangement options were: 'living alone', 'with spouse', 'with spouse and children', 'with children without a spouse', 'with parents and with others'. Ability to work was also self-reported using the question: "During the past 12 months, what was your main activity?" Response options were: 1. working at a paid job or self-employed or going to school, 2. unemployed, 3. caregiving for children, parental leave or caregiving other than children, 4. household work or retired, or 5. long-term illness.

Respondents' health behaviours were examined by asking "In general, would you say that your eating habits are. . .", with options ranging from excellent to poor. These were dichotomized by grouping excellent, very good and good into a 'good' category and fair and poor into the 'poor' category. Smoking status was dichotomized into 'smoking' vs. 'not smoking' and alcohol consumption categorized into 'every day', '4–6 times per week', '2–3 times per week', 'once weekly', 'once or twice per month', 'not in the past month', or 'never had a drink'.

### Box 1. Resilience questions

Thinking about your life in general, how often would you say you:

1. have enough energy to meet life's challenges

2. have a hopeful view of the future

3. are confident in your abilities, even when faced with challenges

4. are able to admit when you have done something wrong

5. have something to look forward to in life

6. have people you can depend on to help you when you really need it

7. are able to bounce back quickly after hard times

8. learned something from those experiences

9. had a hard time accepting those difficulties and moving on with your life

10. after difficult times, you were able to continue going about your life the way you normally do

Ten questions adapted from the reliable and validated Resilience Brief Scale [37] assessed resilience (see Box 1). Responses used a 5-point Likert scale ranging from Always (= 1) to Never (= 5) with totals ranging from 10–50. Given there is no standardised cut-off for high resilience using this series of questions, a cut-point of one standard deviation below the mean (a score of 15) was set as the upper limit for high resilience with greater than 15 representing low resilience. Responses were only included in the composite variable if all questions were answered (n = 18,867).

Other variables included mental/psychological disability status, importance of spiritual or religious beliefs, and frequency of internet use.

### Statistical analysis

Descriptive statistics were calculated for the whole population and for each SRH/health satisfaction category. The significance of bivariate relationships between explanatory variables and SRH/satisfaction groups was assessed using Chi-square tests. For CRT analysis, those with missing data in the outcome variable (n = 124) were removed from the analysis. This left a study population of 19,485. A training:test (30:70) split sample validation was conducted with a maximum tree depth of 6, a minimum parent node of 100 and a minimum child node of 50.

To quantify the level of disorder in data and selection of homogeneous subsets of data we applied the Gini impurity with a minimal improvement set at 0.0001 and equal cost (the full SPSS syntax is available upon request). To assess the validity of prediction accuracy and to ensure the stability of the generated tree, using the same parameters as above we performed a 10-fold cross-validation. All yielded about 24% misclassification. Since there were essentially no meaningful differences in the patterns of the trees generated in these validation processes, we concluded that 76% correct classifications were sufficient for a reliable and stable tree. All analyses were conducted using SPSS version 27.

### Ethics approval

This study involved only secondary use of a pre-existing government of Canada dataset. Participants consented to involvement in data collection for the GSS and the GSS, itself, had ethics approval. Verbal consent was obtained before data collection.

## Results

We present details of findings here and a summary of their meanings in the discussion. Out of the 19,609 original participants, 124 participants were excluded because of missing data on the outcome variable. We started the analysis with 19,485 data points. Table 1 shows descriptive statistics for the variables used in this study. About 55% of the survey population included were female. 29% older than 65, and 78% lived in urban settings. Notably, 85% and 72% of people surveyed reported SRH and satisfaction with health (SH) as good, respectively. Before combining SRH and SH variables to generate a composite measure for our outcome, we examined their bivariate relationships with sex. Both showed significant associations according to the results of Chi-square test (p. value for SRH <0.001 and for SH = 0.003).Bivariate analysis (Table 1) demonstrated that the two outcome groups differed significantly in the distribution of all explanatory variables, with the exception of sex, in particular, but also population centre indicator, and importance of religion/spirituality.

We next examined congruence between reports of SRH and SH by performing a simple cross-tabulation (Table 2). Most participants (81%) had congruent assessments of SRH and SH, 69% reported good for both measures and 12% reported poor self-assessments of the two indicators. However, a small proportion had either good SRH, but were dissatisfied with their health (16%); or poor SRH, but were satisfied with their current health status (3%).

Guided by Table 2 because there were relatively small numbers of individuals with discrepancies in their reported SRH and SH we concluded that generating a regression tree with four outcome groups, though technically possible, would have been very unstable and hard to interpret and thus excluded from our analysis plan.

The predictors of good SRH/SH identified by the regression tree and their relative importance are shown in Fig 1. Healthy eating was the first splitter and identified as the most important factor for the outcome. Occupation, despite appearing first in the third level of the Tree was pivotal in generating many nodes. The regression tree correctly classified 93.3% of participants with positive outcomes and resulted in 24 terminal nodes, ten of which were deemed important. Correct classification for those with negative outcome was 35% yielding a total correct classification of 76%. We defined important nodes as those subgroups with a frequency of reporting good for both indicators more than 20% *different* from the rate for the whole sample. Since 68.9% of the total population reported good SRH/SH any nodes that reported the rates of this positive outcome larger than 83 or smaller than 55 were deemed *important*. Summary characteristics of these 10 important subgroups are described in Table 3.

Eight important nodes identified the subpopulation with poor SRH/SH with health (between 19.6% and 46.6% reported the positive outcome lower than the whole population rate of 68.9%) in comparison to the whole survey population and two nodes (node 35 and 38) identified subpopulations with reported better SRH/SH than the whole sample (68.9%; node 0, Fig 2). The first five branch points of the tree were: (1) healthy eating; (2) mental/psychological disability; (3) perceived social class; (4) ability to work in the past 12 months; and (5) resilience. The remaining variables identified by the regression tree were measures of socioeconomic status such as education and living arrangement, as well as behaviours such as use of the internet, smoking and alcohol consumption. These were lower down in the tree indicating they were of less importance to ratings of health and satisfaction with health.

**Table 1. Distribution of variables for the whole population, those with good SRH and high SH, and those with poor SRH, SH, or both.**

| Variable | | Number (Column %) | | | p-value* |
|---|---|---|---|---|---|
| | | **All** | **Good SRH and good SH** | **Any poor satisfaction or SRH** | |
| Self-rated health (SRH) | Good | 16,617 (84.7) | N/A | N/A | N/A |
| | Poor | 2,921 (14.9) | N/A | N/A | |
| | Missing | 71 (0.4) | | | |
| Satisfaction with health (SH) | Good | 14,022 (71.5) | N/A | N/A | N/A |
| | Poor | 5,478 (27.9) | N/A | N/A | |
| | Missing | 109 (0.6) | | | |
| Age | = <65 | 13,985 (71.3) | 9,854 (73) | 4,051 (67) | <0.001 |
| | >65 | 5,624 (28.7) | 3,594 (27) | 1,986 (33) | |
| Sex | Male | 8,782 (44.8) | 6,006 (45) | 2,722 (45) | 0.295 |
| | Female | 10,827 (55.2) | 7,442 (55) | 3,315 (55) | |
| Population centre indicator | Large urban population centres | 15,350 (78.3) | 10,457 (78) | 4,792 (79) | 0.016 |
| | Rural areas and small population centres | 3,629 (18.5) | 2,563 (19) | 1,046 (17) | |
| | Prince Edward Island | 630 (3.2) | 428 (3) | 199 (3) | |
| Perceived social class | Upper class | 233 (1.2) | 177 (1) | 55 (1) | <0.001 |
| | Upper-middle class | 3,321 (16.9) | 2,639 (20) | 679 (11) | |
| | Middle class | 12,230 (62.4) | 8,668 (66) | 3,531 (60) | |
| | Lower-middle class | 2,749 (14) | 1,491 (11) | 1,253 (21) | |
| | Lower class | 628 (3.2) | 224 (2) | 399 (7) | |
| | Missing | 448 (2.3) | | | |
| Family income | <$25,000 | 2,567 (13.1) | 1,443 (11) | 1,100 (18) | <0.001 |
| | $25,000–49,999 | 3,826 (19.5) | 2,436 (18) | 1,355 (22) | |
| | $50,000–74,999 | 3,630 (18.5) | 2,449 (18) | 1,162 (19) | |
| | $75,000–99,999 | 2,893 (14.8) | 2,108 (16) | 772 (13) | |
| | $100,000–124,999 | 2,117 (10.8) | 1,552 (12) | 559 (9) | |
| | >$125,000 | 4,576 (23.3) | 3,460 (26) | 1,089 (18) | |
| Education highest level achieved | High School or less | 7,557 (38.5) | 4,876 (36) | 2,652 (44) | <0.001 |
| | College or Trade Diploma | 5,705 (29.1) | 3,935 (29) | 1,756 (29) | |
| | University Bachelor's Degree or Equivalent | 4,328 (22.1) | 3,187 (24) | 1,130 (19) | |
| | University above the Bachelor's level | 1,827 (9.3) | 1,366 (10) | 458 (8) | |
| | Missing | 192 (1.0) | | | |
| Main activity past 12 months | Working or School | 11,616 (59.2) | 8,529 (64) | 3,023 (51) | <0.001 |
| | Unemployed | 305 (1.6) | 179 (1) | 125 (2) | |
| | Caregiving—child or other | 703 (3.6) | 519 (4) | 180 (3) | |
| | Household work or retired | 6078 (31) | 3,916 (30) | 2,121 (36) | |
| | Long-term illness | 625 (3.2) | 114 (1) | 502 (8) | |
| | Missing | 282 (1.4) | | | |

(*Continued*)

**Table 1.** (Continued)

| Variable | | All | Good SRH and good SH | Any poor satisfaction or SRH | p-value* |
|---|---|---|---|---|---|
| | | | Number (Column %) | | p-value* |
| Living arrangement | Alone | 5,462 (27.8) | 3,421 (25) | 1,997 (33) | <0.001 |
| | With spouse | 6,150 (31.4) | 4,313 (32) | 1,802 (30) | |
| | With spouse and children | 4,265 (21.7) | 3,113 (23) | 1,131 (19) | |
| | With children alone | 1,054 (5.4) | 655 (5) | 390 (6) | |
| | With parents | 1,618 (8.3) | 1,215 (9) | 395 (7) | |
| | With others, undefined | 1,060 (5.4) | 731 (5) | 322 (5) | |
| Psychological disability status | Yes | 1,984 (10.1) | 756 (6) | 1,220 (21) | <0.001 |
| | No | 17,330 (88.4) | 12,592 (94) | 4,698 (79) | |
| | Missing | 295 (1.5) | | | |
| Resilience | High | 3,294 (16.8) | 811 (6) | 1,460 (25) | <0.001 |
| | Moderate to Low | 15,573 (79.4) | 12,251 (94) | 4,318 (75) | |
| | Missing | 742 (3.8) | | | |
| Importance of religion and spirituality | Important | 11,714 (59.8) | 8,027 (60) | 3,657 (61) | 0.12 |
| | Not important | 7,637 (38.9) | 5,296 (40) | 2,323 (39) | |
| | Missing | 258 (1.3) | | | |
| Internet use | Yes | 17,196 (87.7) | 12,142 (90) | 4,995 (83) | <0.001 |
| | No | 2,350 (12.0) | 1,292 (10) | 1,033 (17) | |
| | Missing | 63 (0.3) | | | |
| Healthy eating | Good | 16,385 (83.6) | 12,161 (90) | 4,168 (69) | <0.001 |
| | Poor | 3,150 (16.1) | 1,277 (10) | 1,861 (31) | |
| | Missing | 74 (0.4) | | | |
| Smoking | Yes | 3,027 (15.4) | 1,858 (14) | 1,161 (19) | <0.001 |
| | No | 16,521 (84.3) | 11,585 (86) | 4,869 (81) | |
| | Missing | 61 (0.3) | | | |
| Alcohol consumption | Daily | 1,093 (5.6) | 765 (6) | 325 (5) | <0.001 |
| | 4-6x/week | 1,549 (7.9) | 1142 (8) | 404 (7) | |
| | 2-3x/week | 3,127 (15.9) | 2,353 (18) | 769 (13) | |
| | Once weekly | 2,715 (13.8) | 2,017 (15) | 693 (12) | |
| | Once or twice monthly | 4,722 (24.1) | 3,261 (24) | 1,440 (24) | |
| | Not in past month | 3,155 (16.1) | 1,877 (14) | 1,260 (21) | |
| | Never | 3,173 (16.2) | 2,023 (15) | 1,134 (19) | |
| | Missing | 75 (0.4) | | | |

*from Chi-square tests

Two pathways that lead to terminal nodes that identified subgroup with higher than total population rates of good SRH/SH were mostly defined by perceived healthy eating, absence of psychological disability, middle social class, and high resilience (nodes 35 and 38). Other factors such as smoking and internet use also had some, although limited predictive impacts.

Perception of dietary health emerged as an important predictor. Six (out of eight) subgroups with lower perceived health reported poor diet. Subgroups with good diet but lower

**Table 2. Cross-tabulation of self-rated health and satisfaction with health.** Greyed categories were grouped for the CRT analysis.

|  |  | Self-rated health (SRH) | | Total |
|---|---|---|---|---|
|  |  | **Good** | **Poor** |  |
| **Satisfaction with health** | Good | 13,448 (69%) | 569 (3%) | 14,017 (72%) |
|  | Poor | 3,133 (16%) | 2,335 (12%) | 5,468 (28%) |
|  | Total | 16,581 (85%) | 2,904 (15%) | 19,485 |

SRH/SH suffered from other health issues such as mental disability, inability to work due to illness, and low resilience (nodes 10 and 28). Notably, the characteristics of limited statistical importance, that is, of limited predictive value, were sex, population centre indicator, smoking, and religion/spirituality activities.

## Discussion

We found that most Canadians surveyed had good SRH and congruent satisfaction with their health. However, the approximately 30% who reported either poor SRH and/or poor SH could benefit from interventions that improve health and well-being. This study is the first to identify predictors of SRH/SH considered together and in a general population. Others have used ordinary multivariate regression analyses to examine the independent importance of medical, behavioural, social and economic variables [3, 38–40] in the perception of health. In view of the complexity of relationships among characteristics that shape SRH use of simple regression has been criticized [41] regression tree techniques are a means of better exploring this non-linear complexity [42].

Combining SRH and SH has the potential to explain inconsistencies and lack of reproducibility of SRH when subgroups such as women and men are considered, and to demonstrate the complex interplay of multiple health-related, behavioural and social characteristics. A comprehensive review of sex differences in all studies of SRH is well beyond the scope of this paper. In general, though, findings of differences in men's and women's SRH have been

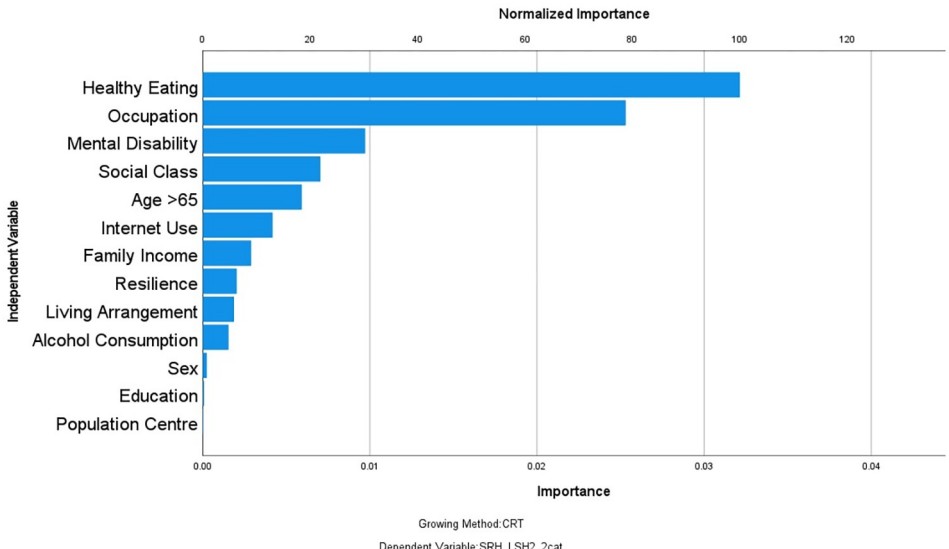

**Fig 1. Variable importance to the model.** Assigned importance and normalized importance to the model for each variable included in the final CRT model.

**Table 3. Description of important subgroups identified by CRT.**

| Subgroup/ Node number | Number in the Subgroup | % good | Node characteristics |
|---|---|---|---|
| 10 Lower | 368 | 31.5 | Good diet, mental disability, no work due to illness or retirement |
| 13 Lower | 393 | 36.6 | Poor diet, lower or upper social class, no work due to illness or retirement |
| 14 Lower | 241 | 21.2 | Poor diet, lower or upper social class, working or school |
| 21 Lower | 280 | 32.1 | Poor diet, middle social class, no work due to illness or retirement, living alone or with spouse and children |
| 22 Lower | 173 | 38.7 | Poor diet, middle social class, no work due to illness or retirement, living alone or with spouse or parents |
| 24 Lower | 173 | 26 | Poor diet, middle social class, working or school, mental disability |
| 28 Lower | 143 | 19.6 | Good diet, no mental disability, moderate/low resilience, not working due to illness |
| 35 Higher | 1339 | 90.4 | Good diet, no mental disability, high resilience, middle social class, not smoking, use of internet |
| 38 Higher | 108 | 88.9 | Good diet, no mental disability, high resilience, middle social class, smoking, living with spouse or children or parents |
| 43 Lower | 236 | 46.6 | Poor diet, middle social class, working or school, no mental disability, never to moderate alcohol consumption, living with spouse and/or children |

Only nodes that identified subgroups with a rate of good SRH/HS 20% larger or smaller than the total population baseline (68.9%) are reported

consistent but also confusing in two particular ways. First, while sex differences in SRH are the norm, uniformity in which group reports greater subjective health varies [1, 6, 8]. Using GSS data there were significant differences in bivariate analyses of sex and SRH, alone, (data not reported) as there were with sex and SH. However, sex was not significantly related to the outcome of SRH combined with SH in bivariate analysis (p = 0.295) nor was it identified in the regression tree as a predictor. As a sensitivity analysis we also constructed sex-stratified regression trees. Results were similar to those for the whole sample (data not shown). The lack of significance of sex as a predictor of the combined outcome of SRH/SH suggests that adding satisfaction with health may correct for sex/gender differences in interpretation of the meaning and, hence, rating of subjective health. The second way in which adding SH to SRH may deepen meaning has to do with women's frequent although not universal reports of poorer SRH relative to men, alongside their greater longevity [10–13]. This paradox also raises questions as to the meanings men and women attribute to the measure, SRH [43]. In keeping with the findings of others, perceived healthy eating [38, 39], absence of mental health issues [44], and ability to maintain a meaningful occupation were the strongest predictors of good SRH and good satisfaction with health. Others have demonstrated that perceptions of healthy eating are strongly associated with socioeconomic status [45, 46], suggesting that this first branch in our CRT analysis might be a proxy measure for socioeconomic effects. This is reinforced by the finding that employment can overcome perceived poor eating habits and tip the balance toward reporting of good SRH/SH (node 12).

Also described by others, resilience, or the ability to overcome life's challenges and thrive emerged as an important predictor of good SRH/SH [47]; however, this effect was more nuanced than has been previously noted. High resilience is responsible for both important

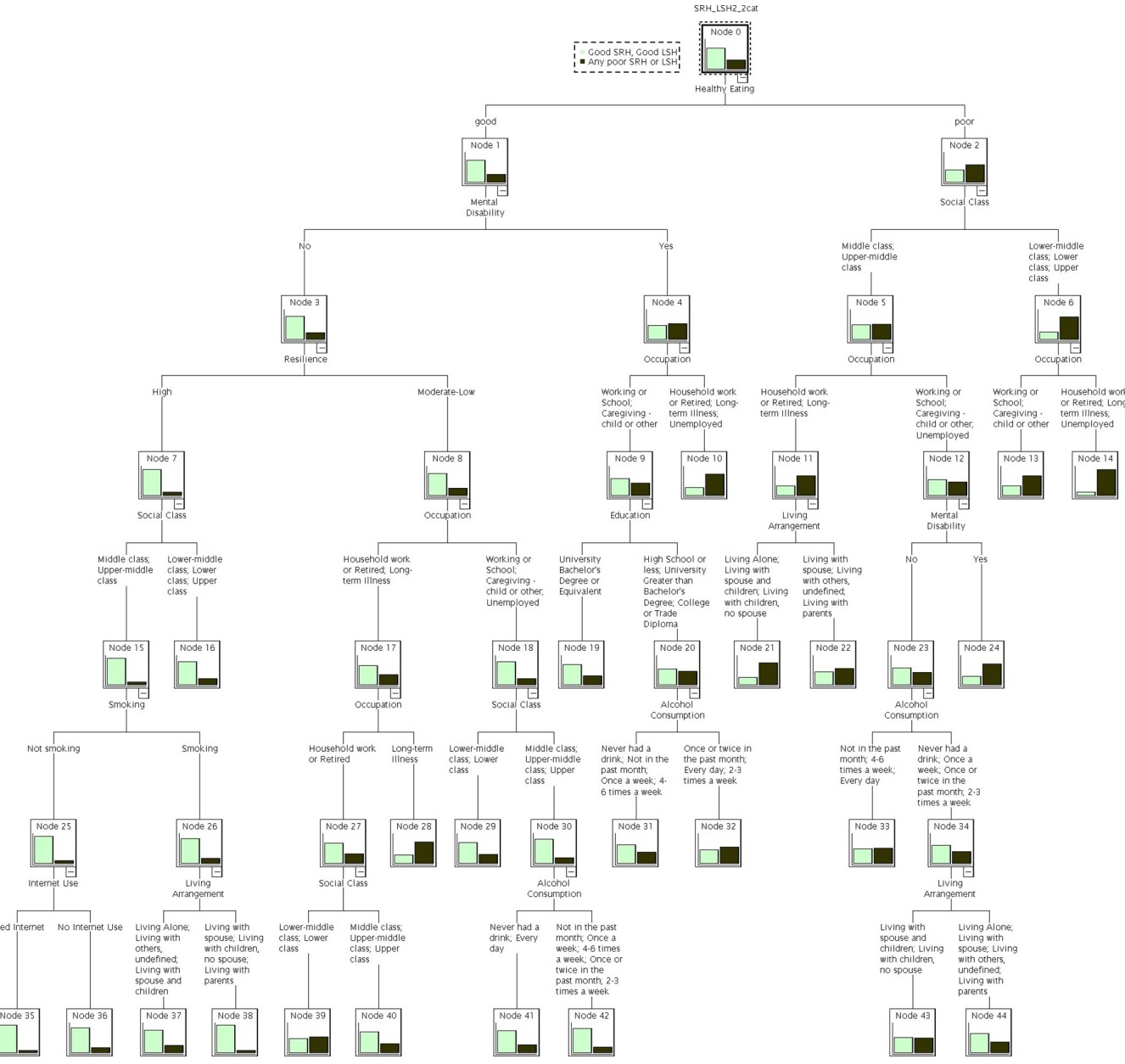

**Fig 2. Classification regression tree analysis.** Outcome is defined as good SRH/SH vs. any poor rating of SRH or SH. Maximum number of branches capped at 6. Minimum parent node size of 100. Minimum child node size of 50.

subgroups that identified subgroups with high rates of good SRH/SH (nodes 35 and 38). Even with low-moderate resilience, we found that participants might overcome other predictors of poor SRH/SH by having an active occupation or a perceived social status greater than middle class (node 30). Consistent with the literature [40], this suggests that predictors of poor SRH/SH might be modified by one's socioeconomic status. Another related social factor that we identified was occupation. With a similar moderate and low level of resilience those with an active occupation showed 80% rates of good SRH/SH (node 18) whereas only 20% of those with no occupation secondary to illness rated their health/satisfaction as good (node 28).

Behavioural variables such as smoking, alcohol consumption and internet use were identified further down in the regression tree. Due to their limited importance they will not be discussed here to avoid potential over-interpretation of an exploratory analytic method.

## Strengths and limitations

We adopted a new methodology to explore the relationships between predictors of the outcome that are not easily observable using traditional regression methods. Our study was innovative because it focused on the combined outcomes of self-rated health and satisfaction with health. The GSS is nationally representative, but cross-sectional, which precludes any interpretation of temporality. Furthermore, data available were limited to the questions included. Respondents were not asked specifically about gender identity but only to select whether they were male or female. To compensate for lack of medical data we considered using available measures of physical health such as disability or chronic pain, but these variables' high levels of missing data precluded this analysis. We did see a signal of physical health in the 'activity' variable that suggested inability to work due to long-term illness had an impact on SRH/SH. Nevertheless, the data availability issue is another potential limitation since higher mortality rates have been observed in individuals who report poor SRH relative to those with incongruent self-rated and objective health status [48].

Finally, the results of CRT analysis are exploratory. Future studies should use longitudinal data, including objective physical health measures, and causal mediation to confirm results.

## Conclusions

Sex differences in subjective health ratings disappear when measures of SRH and SH are combined. This combination may capture day to day experience along with medical circumstances and produce a more comprehensive picture of well-being. SRH/SH seems to correct for varying interpretations of the meaning of self-rated health, alone. The disappearance of sex differences suggests that components of men's and women's different reporting of subjective health may be an artefact of definition. We do acknowledge that despite the intersectional analysis inherent in decision tree designs there are almost certainly some gender differences (that is, intersections of sex and social circumstances) that predict SRH/SH and have gone unmeasured in this study. Our findings were that the interplay of physical health, mental health, behaviour and socio-economic status, but not sex, shape perceived health and satisfaction with it. Particularly diet, resilience, and ability to work and cope with life stressors were strong predictors of good health and satisfaction with it. There is growing evidence that policies requiring limited political or economic upheaval can foster resilience among adults [49]. CRT analysis allowed us to identify complex, non-linear relationships that would not have emerged using classical multivariable regression analyses. Future studies of SRH and SH could continue exploring these nuanced, intersectional relationships to better guide public health policies and avoid putting individuals into an 'all or none' basket, that is, avoid assuming homogeneity within groups such as women or men.

## Supporting information

**S1 File. Frequency distributions of original scales of SRH and SH.**
(DOCX)

## Acknowledgments

We acknowledge the participants of the GSS survey, without whom this research would not be possible.

## Author Contributions

**Conceptualization:** Afshin Vafaei, Jocelyn M. Stewart, Susan P. Phillips.

**Data curation:** Jocelyn M. Stewart, Susan P. Phillips.

**Formal analysis:** Afshin Vafaei, Jocelyn M. Stewart.

**Funding acquisition:** Susan P. Phillips.

**Investigation:** Susan P. Phillips.

**Methodology:** Afshin Vafaei, Jocelyn M. Stewart, Susan P. Phillips.

**Project administration:** Susan P. Phillips.

**Writing – original draft:** Afshin Vafaei, Jocelyn M. Stewart, Susan P. Phillips.

**Writing – review & editing:** Afshin Vafaei, Jocelyn M. Stewart, Susan P. Phillips.

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
