## [Decision Letter · Decision Letter 0]

26 May 2023

PONE-D-23-10826Descriptive regression tree analysis of intersecting predictors of adult self-rated health: does gender matter? A cross-sectional study of Canadian adultsPLOS ONE

Dear Dr. Vafaei,

Thank you for submitting your manuscript to PLOS ONE. After careful consideration, we feel that it has merit but does not fully meet PLOS ONE’s publication criteria as it currently stands. Therefore, we invite you to submit a revised version of the manuscript that addresses the points raised during the review process.

The three reviewers have done a meticulous job, and propose several modifications that will significantly improve the quality of your work.

We look forward to receiving your revised manuscript.

Kind regards,

Jordi Gumà, Ph.D.

Academic Editor

PLOS ONE

Journal Requirements:

“AV was supported by The Canadian Institutes of Health Research, award number 161787.”

Reviewers' comments:

Reviewer's Responses to Questions

**Comments to the Author**

1. Is the manuscript technically sound, and do the data support the conclusions?

Reviewer #1: Yes

Reviewer #2: Yes

Reviewer #3: Partly

2. Has the statistical analysis been performed appropriately and rigorously? 

Reviewer #1: Yes

Reviewer #2: I Don't Know

Reviewer #3: Yes

3. Have the authors made all data underlying the findings in their manuscript fully available?

Reviewer #1: Yes

Reviewer #2: Yes

Reviewer #3: Yes

4. Is the manuscript presented in an intelligible fashion and written in standard English?

Reviewer #1: Yes

Reviewer #2: Yes

Reviewer #3: Yes

5. Review Comments to the Author

Reviewer #1: The research topic is interesting and has a series of implications for a principled discussion in public health, this is in light of the importance of using survey data for decision-making by policy-makers in the health system.

The authors of the study make an interesting and even original (to some extent) use of the classification and regression tree (CRT) for a deeper understanding of the research issue.

However - there are a number of aspects that research editors must consider in relation to them and treat them with this seriousness. I will detail:

1. The researchers are required to go deeper and detail the line of research questions that are the basis of the current study, and to explain the scientific rationale behind each of them.

2. The researchers state that "our methodology was intentionally chosen to make no assumptions about findings". This approach is at odds with a precise scientific investigation in which the researchers present a series of research hypotheses and try to test them. Therefore, the researchers are required to explain in depth why they decided to take this approach.

3. Also, it is proper and accurate for the researchers to formulate a series of research hypotheses which are mainly based on scientific theory, and examine them within the framework of the research - deepening and discussing them in the discussion and summary chapter.

4. The scientific need that led the researchers to promote the current study should be expanded and detailed more. What do they think is missing in the field and why is it important to advance the examination of the research questions.

5. The scientific description and explanation about the regression trees should be deepened and expanded, and explain why this method was chosen as the most appropriate and accurate estimation method in the current study.

6. There is no scientific presentation regarding the considerations that guided the researchers in choosing the explanatory variables that were included - why these are the most suitable variables and how they are supposed to promote the degree of understanding and the research goals.

7. The discussion is very sparse and not in-depth enough. Its deepening and expansion is required in accordance with all the aforementioned issues.

8. The research conclusions are not comprehensive enough. A rewriting of them is required, which takes into account the various findings as well as all the aspects and extensions which, as mentioned, need to be deepened and addressed.

I will conclude by saying that although the subject of the research is valuable and conceptually important to policy makers in the health system, in order for it to be - it is necessary to deepen and develop what is said in the research in a thorough and thorough manner, on the basis of all the comments mentioned above.

Reviewer #2: The study investigates the combined measure of SRH and SH using classification and regression tree (CRT) analyses. It was an interesting topic and the manuscript was well-written. However, I have some comments that could improve the manuscript.

[Background]

It would be good to see a reference for this sentence (As an example, it is possible to determine whether different combinations of SES, resilience and ability to work shape SRH and SH differently.)

[Methods]

- Are there any previous research that showed the effects of combining SRH and SH measures or is this the authors' first attempt to do so? Although it may be a practical and effective way to measure subjective health, it would be good to have some evidence for the measurement. Are there any references or evidence for the cut-off points for SRH and SH that were used in this study?

- Also, It would help readers understand more about methodology if the authors describe more in detail in the Methods section.

- The authors stated that they chose to make no assumptions, but it seems that this statement shows the direction of the assumptions or the purpose of the study.(->We propose that individuals with dissatisfaction with their health and/or poor SRH could benefit from interventions aimed at improving these factors and therefore the latter three categories described were grouped for the analysis.) Maybe it would be clearer to state the purpose of the study in the last section of the Background.

- I wonder if there are no differences among groups that were classified as the same category (i.e., low SRH-high HS, high SRH-low SH and low SRH-low SH). Can you explain more about categorizing the main outcome?

[Results]

- The authors mentioned that it was the first study to focus on the general population. Characteristics of the participants should be presented more in the Results.

- Could you enhance the quality of images, especially Figure 2? Currently, it is difficult to read the words of the images.

[Discussion]

- The title of the study implies that the study focused on examining gender differences. However, it seems that the differences in the concept 'sex' and 'gender' were not considered in this study. As I read the manuscript, it seemed that 'sex' refers to the biological differences between males and females in this study. Although the distinction between sex and gender may not be easy, I wanted to know how the participants answered their sex/gender.

Moreover, the results that sex was not a significant predictor may not mean there were no gender differences. It could be related to population characteristics or limitations of the study datasets. It would be good to add deeper discussions regarding the implications of these results.

Reviewer #3: Dear Authors

I enjoyed reading your manuscript. I think it is a relevant topic for the field. I feel the idea of combining self-rated health with health satisfaction is very interesting and may help overcome the problems of using only self-rated health. I also feel using exploratory classification tree analysis here is a good technique to find subgroups otherwise not identified.

However, I have a few comments and would appreciate if these could be addressed in a revision process.

1. Major point: In the discussion of the paper the following part is included “Of note, sex did not emerge as a predictor of the combined outcome of SRH/SH suggesting that adding satisfaction with health corrected for gender differences in interpretation of the meaning of SRH, alone.” I understand from the background section of the paper, that in previous research sex appeared as a predictor of SRH. However, what I asked myself: is sex a predictor for SRH in this particular population? Did you perform a bivariate analysis for sex and SRH? Was this significant? Did you perform a CRT analysis for only the outcome SRH? Did sex appear as an important variable in this analysis? I feel like adding the results of these analyses to the paper would elevate the manuscript significantly. What if sex was never a predictor of SRH in this population from the beginning. Then the statement in the discussion that “adding satisfaction with health corrected for gender differences” cannot be made.

2. Interchangeable use of the terms “sex” and “gender”. I noticed that throughout your manuscript, you used the terms “sex” and “gender” interchangeable. However, these two terms do not describe the same concept. It should be clearly defined whether sex or gender is addressed. There is also the possibility of using the term “sex/gender” throughout the manuscript to acknowledge the fact that the two are intertwined as expressed in the embodiment theory. However, if this term is used it has to be explained and described. (See e.g. Hammarström & Annandale 2012 https://doi.org/10.1371/journal.pone.0034193 and Springer et al. 2012 (https://doi.org/10.1016/j.socscimed.2011.05.033)

3. Dichotomization of outcome variables: I understand that it makes sense to dichotomize the outcome variables for CRT analysis. However, I would appreciate a better explanation of how it was decided to dichotomize at the chosen cut-off points. E.g. was the cut-off for SRH between good and fair chosen before the distribution of the data was seen, or afterwards based on the distribution. I would actually find it helpful to see the initial distribution too, eg. Did the majority of people categorized as good health actually indicate ‘excellent’, ‘very good’, ‘good’ health. Same for the outcome satisfaction with health, here it is stated that “This cut-point was selected given the apparent bimodal distribution of responses.” Is it possible to show this bimodal distribution (maybe in a supplemental material).

4. As sex (or gender) is one of the main focusses of the paper (at least based on the title of the paper) I would expect a clearer explanation of how this variable was measured. It is stated that “Sex was dichotomous (male/female) and based on participant's self-identification.” Does this mean that participants could identify as only male or female in the questionnaire (forced choice), or did they have the opportunity to identify as divers, trans, intersex or another category and this was later dichotomized in any way?

5. CRT analysis: Why was the maximum tree depth chosen to be 6? Is this based on any prior literature or recommendations? Or were indeed different depths explored and this was the one with the lowest misclassification error? I feel like a depth of 6 is rather high. In other papers performing CRT analysis I have seen trees with less depth in order to retain interpretability.

6. CRT analysis: Can some more information be provided on the splitting algorithm used by this particular CRT analysis. There are many different types of classification and regression tree analyses, and I could not clearly understand which algorithm was used (e.g. misclassification error, information gain or Gini impurity).

7. CRT analysis: It is stated that “75% correct classifications were sufficient for a reliable and stable tree”. Is there any literature that can suggests that this percentage of correct classifications is acceptable?

8. Description of results: I am not sure if I understand the following sentence “The regression tree correctly classified almost 90% of participants with positive outcomes and resulted in 24 terminal nodes, ten of which were deemed important.” This means that 90% of participants with good SRH/satisfaction were classified correctly, but from the participants with a bad SRH/satisfaction a lower percentage were classified correctly? Could this percentage also be reported?

9. Description of results: Minor point: I had to read the sentence “…more than 20% different from the whole sample rate that was estimated equal to 68.9%” a few times till I understood. From the construction of the sentence, I thought there is a sample rate of 68.9%, until I realized that within the whole sample 68.9% rated their SRH/satisfaction as good. Maybe the sentence can be constructed differently.

10. Description of results: I was a bit confused by the following two sentences “Eight nodes enriched for poor SRH/satisfaction with health (between 19.6% and 46.6%) in comparison to the whole survey population. In this context, enriched indicates that the proportion of respondents with positive outcomes (good SRH and high SH) in the specified subpopulation

exceeds that among the whole study population (68.9%; node 0, Figure 2)”. The wording “in this context” indicated for me that in both sentences the same kind of enrichment is described, but in fact in the first sentence the nodes with a lower SRH/satisfaction are described but in the second sentence does nodes with a higher SRH/satisfaction.

11. Table 1: Could it be that for the age variable the signs < and > were mixed up? Or where only 28.7 % of the population younger than 65 years?

12. Table 1: I was confused by the fact that according to Table 1 amongst people with poor satisfaction or SRH a higher percentage of people (25%) showed a high resilience compared to the 6% high resilience amongst people with a good SRH and good SH. I would have expected the opposite. Could this be discussed?

13. Table 1: Similar for alcohol consumption. According to Table 1 people with poor satisfaction or SRH drink less alcohol than people with a good SRH and good SH. I would have expected the opposite. Could this be discussed?

14. Table 1: Minor point: Sometimes it is not clear in which line certain numbers belong. One example: in the row “Perceived social class”, do the numbers 224 (2) 399 (7) belong to Lower class or Missing.

15. In the conclusion of the paper nothing is mentioned about gender (or sex). I found that to be incoherent with both the title of the paper “does gender matter?” and the conclusion part of the abstract “Combining SRH and SH eliminated sex as a predictor of SRH/SH,…”

6. PLOS authors have the option to publish the peer review history of their article (what does this mean?). If published, this will include your full peer review and any attached files.

Reviewer #1: No

Reviewer #2: No

Reviewer #3: **Yes: **Lisa Dandolo

---

## [Author Response · Author response to Decision Letter 0]

14 Aug 2023

Review Comments to the Author

Reviewer #1: The research topic is interesting and has a series of implications for a principled discussion in public health, this is in light of the importance of using survey data for decision-making by policy-makers in the health system.

The authors of the study make an interesting and even original (to some extent) use of the classification and regression tree (CRT) for a deeper understanding of the research issue.

However - there are a number of aspects that research editors must consider in relation to them and treat them with this seriousness. I will detail:

1. The researchers are required to go deeper and detail the line of research questions that are the basis of the current study, and to explain the scientific rationale behind each of them.

Response: The rationale for our research is summarized in pages 3 to 5. Briefly, we conceptualized, following the health asset approach to health promotion (ref# 17), that adding satisfaction with health (SH) to SRH provides a more comprehensive measure of subjective health and well-being. Our aim was to identify the characteristics of those enjoying high levels of SRH and SH, as specified in page 5. 

2. The researchers state that "our methodology was intentionally chosen to make no assumptions about findings". This approach is at odds with a precise scientific investigation in which the researchers present a series of research hypotheses and try to test them. Therefore, the researchers are required to explain in depth why they decided to take this approach.

Response: Thank you for the thoughtful comment. We agree that this statement is misleading. Classification and regression Tree (CRT), being generally a machine learning technique for exploratory analysis is free of assumptions of traditional regression methods, and more importantly is not a hypothesis testing procedure. We revised the statement (page 5). 

3. Also, it is proper and accurate for the researchers to formulate a series of research hypotheses which are mainly based on scientific theory, and examine them within the framework of the research - deepening and discussing them in the discussion and summary chapter.

Response: We hope our response to comment 1 and 2 addresses this concern as well. 

4. The scientific need that led the researchers to promote the current study should be expanded and detailed more. What do they think is missing in the field and why is it important to advance the examination of the research questions.

Response: The main scientific contribution of this study is its focus on combining SH and SRH as a measure of general health and well-being. To the best of our knowledge there is no other study that identifies characteristics of individuals with high levels of SRH/SH and thus can “provide a broader and deeper picture of characteristics of those with highest reported current health and well-being, characteristics that augment health assets and that can be fostered, page 4). Additionally, as stated in the revised paragraph (page 5), these results can guide “policies and practices that extend beyond medical prevention and treatment”. We modified the statements in pages 4 and 5 to provide clearer information. 

5. The scientific description and explanation about the regression trees should be deepened and expanded, and explain why this method was chosen as the most appropriate and accurate estimation method in the current study.

Response: Please see our response to comment # 2, we also edited our description of CRT in the last two paragraphs of the background for clarity 

6. There is no scientific presentation regarding the considerations that guided the researchers in choosing the explanatory variables that were included - why these are the most suitable variables and how they are supposed to promote the degree of understanding and the research goals.

Response: As stated in the response to comment # 4 there was no study to specifically provide evidence on predictors of combinations of high SRH and high SH. We chose the explanatory variables based on the main social and demographic determinants of health, the literature on predictors of SRH (reference 1-20....) adding a validated measure of resilience based on the health asset approach to well-being and references 25 to 27. 

7. The discussion is very sparse and not in-depth enough. Its deepening and expansion is required in accordance with all the aforementioned issues.

Response: We have expanded the discussion, particularly with respect to sex and gender and the merit of adding SH to SRH to better measure subjective wellbeing.

8. The research conclusions are not comprehensive enough. A rewriting of them is required, which takes into account the various findings as well as all the aspects and extensions which, as mentioned, need to be deepened and addressed.

Response: The conclusion has been revised to be more comprehensive and, in particular, to highlight the findings around sex/gender.

I will conclude by saying that although the subject of the research is valuable and conceptually important to policy makers in the health system, in order for it to be - it is necessary to deepen and develop what is said in the research in a thorough and thorough manner, on the basis of all the comments mentioned above.

Response: thank you, we hope by adding the clarification notes to the background and rewriting the discussion the merit of the study is clearer now. 

Reviewer #2: The study investigates the combined measure of SRH and SH using classification and regression tree (CRT) analyses. It was an interesting topic and the manuscript was well-written. However, I have some comments that could improve the manuscript.

[Background]

It would be good to see a reference for this sentence (As an example, it is possible to determine whether different combinations of SES, resilience and ability to work shape SRH and SH differently.)

Response: As stated in responses to reviewer 1 and in the manuscript there was no study that identify the predictors of combined SRH and SH. This is the main rationale for this analysis and the main contribution of the study. 

[Methods]

- Are there any previous research that showed the effects of combining SRH and SH measures or is this the authors' first attempt to do so? Although it may be a practical and effective way to measure subjective health, it would be good to have some evidence for the measurement. 

Response: As noted above we were not able to locate any such studies. SRH is a validated measure of subjective health and also a predictor of mortality (e.g., references 3 to 11) and there is anecdotal evidence showing SH is also a potential predictor of health (e.g., reference # 14). These are documented in the background. 

- Are there any references or evidence for the cut-off points for SRH and SH that were used in this study?

Response: Dichotomizing SRH into ‘good’ and ‘fair’ is a common practice in the literature and we added a few references to support our approach. There is no established cut-off point for SH; however, our cut-off point of 7 was not completely arbitrary and selected according to the distribution of responses in our data to provide efficiency in statistical analysis. 

- Also, It would help readers understand more about methodology if the authors describe more in detail in the Methods section.

Response: For clarity, we revised parts of the methods section as well as description of the CRT methodology in the background. 

- The authors stated that they chose to make no assumptions, but it seems that this statement shows the direction of the assumptions or the purpose of the study.(->We propose that individuals with dissatisfaction with their health and/or poor SRH could benefit from interventions aimed at improving these factors and therefore the latter three categories described were grouped for the analysis.) Maybe it would be clearer to state the purpose of the study in the last section of the Background.

Response: Thank you, this concern was also brought up by reviewer #1. You have shown us that we were not clear in the use of the word, assumption, hence we rewrote the whole sentence and elaborated on the rationale and aims of the study, please see page 5. 

- I wonder if there are no differences among groups that were classified as the same category (i.e., low SRH-high HS, high SRH-low SH and low SRH-low SH). Can you explain more about categorizing the main outcome?

Response: There were relatively small numbers of individuals with discrepancies in their reported SRH and SH (please see Table 2). Therefore, generating a Regression Tree with 4 outcome groups, though technically possible, would have been very unstable and hard to interpret. We added a short note to page 12 of the manuscript. 

Another reason for dichotomizing the outcome variable and grouping all with a poor rating of either SRH or SH was purely conceptual: “We propose that individuals with dissatisfaction with their health and/or poor SRH could benefit from interventions aimed at improving these factors and therefore the latter three categories described were grouped for the analysis, page 7).

[Results]

- The authors mentioned that it was the first study to focus on the general population. Characteristics of the participants should be presented more in the Results.

Response: The socio-demographic details of the population are shown in table 1. We have added some socio-demographic statistics to the manuscript (page 11). 

- Could you enhance the quality of images, especially Figure 2? Currently, it is difficult to read the words of the images.

Response: We created figures with resolution guided by the Journal. However, we do observe that the figures converted into PDF are not clear. If you download the original figures using the link on the top of the page they will be much more readable. 

[Discussion]

- The title of the study implies that the study focused on examining gender differences. However, it seems that the differences in the concept 'sex' and 'gender' were not considered in this study. As I read the manuscript, it seemed that 'sex' refers to the biological differences between males and females in this study. Although the distinction between sex and gender may not be easy, I wanted to know how the participants answered their sex/gender.

Moreover, the results that sex was not a significant predictor may not mean there were no gender differences. It could be related to population characteristics or limitations of the study datasets. It would be good to add deeper discussions regarding the implications of these results.

Response: The concerns about sex versus gender has been brought up by other reviewers and we agree that this should be explained better. We have added more notes about sex and gender to the background, discussion and the conclusion.

Reviewer #3: Dear Authors

Reviewer 3: 

I enjoyed reading your manuscript. I think it is a relevant topic for the field. I feel the idea of combining self-rated health with health satisfaction is very interesting and may help overcome the problems of using only self-rated health. I also feel using exploratory classification tree analysis here is a good technique to find subgroups otherwise not identified.

However, I have a few comments and would appreciate if these could be addressed in a revision process.

1. Major point: In the discussion of the paper the following part is included “Of note, sex did not emerge as a predictor of the combined outcome of SRH/SH suggesting that adding satisfaction with health corrected for gender differences in interpretation of the meaning of SRH, alone.” I understand from the background section of the paper, that in previous research sex appeared as a predictor of SRH. However, what I asked myself: is sex a predictor for SRH in this particular population? Did you perform a bivariate analysis for sex and SRH? 

Response: We did perform these analyses– bivariate analyses of sex and SRH, and sex and SH both showed that sex was significant – see expanded section in discussion.

Did you perform a CRT analysis for only the outcome SRH? 

Response: We did not do this analysis because our research question was neither about SRH nor SH, alone, and as conceptualized in the manuscript (also see our response to reviewer # 1) focused on the combination of SRH and SH.

Did sex appear as an important variable in this analysis? I feel like adding the results of these analyses to the paper would elevate the manuscript significantly. What if sex was never a predictor of SRH in this population from the beginning. Then the statement in the discussion that “adding satisfaction with health corrected for gender differences” cannot be made.

Response: We agree that this is a key consideration. We did these bivariate analyses but had not reported them. We have added this info with our interpretation of findings – in the discussion

2. Interchangeable use of the terms “sex” and “gender”. I noticed that throughout your manuscript, you used the terms “sex” and “gender” interchangeable. However, these two terms do not describe the same concept. It should be clearly defined whether sex or gender is addressed. There is also the possibility of using the term “sex/gender” throughout the manuscript to acknowledge the fact that the two are intertwined as expressed in the embodiment theory. However, if this term is used it has to be explained and described. (See e.g. Hammarström & Annandale 2012 https://doi.org/10.1371/journal.pone.0034193 and Springer et al. 2012 (https://doi.org/10.1016/j.socscimed.2011.05.033)

Response: We are embarrassed to say that we failed to follow our own advice to authors to always define sex, gender and sex/gender. These are now defined in the background and we have gone through the manuscript to make sure we mean ‘sex’ when we use that term, and the same for gender and sex/gender. Thank you for these two related references, we added both to our references list

3. Dichotomization of outcome variables: I understand that it makes sense to dichotomize the outcome variables for CRT analysis. However, I would appreciate a better explanation of how it was decided to dichotomize at the chosen cut-off points. E.g. was the cut-off for SRH between good and fair chosen before the distribution of the data was seen, or afterwards based on the distribution. I would actually find it helpful to see the initial distribution too, eg. Did the majority of people categorized as good health actually indicate ‘excellent’, ‘very good’, ‘good’ health. Same for the outcome satisfaction with health, here it is stated that “This cut-point was selected given the apparent bimodal distribution of responses.” Is it possible to show this bimodal distribution (maybe in a supplemental material).

Response: Please see the histogram below. The left skewness of the distribution was expected. Since we observed two peaks at the levels of 5 and 7, we decided to set the cut-off point at 7. As mentioned in the manuscript and also in the response to reviewer 2 there is no established cut-off point to follow and we selected 7 as the best potential cut-off. 

4. As sex (or gender) is one of the main focusses of the paper (at least based on the title of the paper) I would expect a clearer explanation of how this variable was measured. It is stated that “Sex was dichotomous (male/female) and based on participant's self-identification.” Does this mean that participants could identify as only male or female in the questionnaire (forced choice), or did they have the opportunity to identify as divers, trans, intersex or another category and this was later dichotomized in any way?

Response: As with many datasets, there were only two options offered. The GSS did not allow for a response re gender identity. Info re this as a limitation has been added in the limitations section of the discussion.

5. CRT analysis: Why was the maximum tree depth chosen to be 6? Is this based on any prior literature or recommendations? Or were indeed different depths explored and this was the one with the lowest misclassification error? I feel like a depth of 6 is rather high. In other papers performing CRT analysis I have seen trees with less depth in order to retain interpretability.

Response: We had a relatively large sample size and even at the depth of 6, size of most child nodes were larger than 200 (a more comprehensive Tree including all node related statistics is available upon request). Since it was an exploratory inquiry we thought adding more factors into the Tree allows us to investigate the combination effects of more factors and provides a better picture of a complex and multifactorial outcome. 

6. CRT analysis: Can some more information be provided on the splitting algorithm used by this particular CRT analysis. There are many different types of classification and regression tree analyses, and I could not clearly understand which algorithm was used (e.g. misclassification error, information gain or Gini impurity).

Response: We totally agree that there are many algorithms and various options to generate the Tree. Again, our aim was an exploration of potentially modifiable factors for devising interventions and policy goals (see page 4). We also decided not to include technical aspect of the algorithm considering the audience of the Journal. The misclassification errors are reported in the ‘methods’ and we applied the Gini impurity with a minimal improvement set at 0.0001 and equal cost (the full SPSS syntax is available upon request). 

7. CRT analysis: It is stated that “75% correct classifications were sufficient for a reliable and stable tree”. Is there any literature that can suggests that this percentage of correct classifications is acceptable?

Response: We could not find any literature on acceptable levels of misclassification and think it depends on the data and research question. We appreciate any additional information from the reviewer in this regard. 

8. Description of results: I am not sure if I understand the following sentence “The regression tree correctly classified almost 90% of participants with positive outcomes and resulted in 24 terminal nodes, ten of which were deemed important.” This means that 90% of participants with good SRH/satisfaction were classified correctly, but from the participants with a bad SRH/satisfaction a lower percentage were classified correctly? Could this percentage also be reported?

Response: Correct, in fact the exact value of for correct classification was 92.9%, the rate of misclassification for participants with poor SRH/SH was 34.1% and the exact value of total ‘correct’ misclassification was 75.7% as reported in the ‘methods’. 

9. Description of results: Minor point: I had to read the sentence “…more than 20% different from the whole sample rate that was estimated equal to 68.9%” a few times till I understood. From the construction of the sentence, I thought there is a sample rate of 68.9%, until I realized that within the whole sample 68.9% rated their SRH/satisfaction as good. Maybe the sentence can be constructed differently.

Response: Yes, the rate of good SRH/SH in the whole population was 68.9% and we defined important nodes (listed in Table 3) based on differences between rate of the outcome in the subpopulation identified by the ‘node’ and the rate in the whole sample. We edited the sentence slightly and hope it is clearer now.

10. Description of results: I was a bit confused by the following two sentences “Eight nodes enriched for poor SRH/satisfaction with health (between 19.6% and 46.6%) in comparison to the whole survey population. In this context, enriched indicates that the proportion of respondents with positive outcomes (good SRH and high SH) in the specified subpopulation

exceeds that among the whole study population (68.9%; node 0, Figure 2)”. The wording “in this context” indicated for me that in both sentences the same kind of enrichment is described, but in fact in the first sentence the nodes with a lower SRH/satisfaction are described but in the second sentence does nodes with a higher SRH/satisfaction.

Response: A thoughtful comment and we agree that use of the term ‘enrich’ is confusing and probably misleading. We removed this word from the manuscript rewrote the sentence (page 12). 

11. Table 1: Could it be that for the age variable the signs < and > were mixed up? Or where only 28.7 % of the population younger than 65 years?

Response: Thank you for detecting this error, rectified. 

12. Table 1: I was confused by the fact that according to Table 1 amongst people with poor satisfaction or SRH a higher percentage of people (25%) showed a high resilience compared to the 6% high resilience amongst people with a good SRH and good SH. I would have expected the opposite. Could this be discussed?

Response: Despite higher resilience in poor SRH/SH people, still high resilience was a predictor of good SRH/SH (please see Figure 2 and Table 3). The goal of this exploratory analysis was to identify characteristics of subgroups with the highest probability of low or high rates of the outcome. Thus, we tried to avoid over-interpretation of bivariate results of table 1 in the manuscript. However, higher resilience in groups with poorer health specially in older adults has been observed in other studies (reference # 49) and is consistent with the intersectionality theories that suggest vulnerable groups develop coping mechanisms to overcome adversities (reference # 1 among many others). 

13. Table 1: Similar for alcohol consumption. According to Table 1 people with poor satisfaction or SRH drink less alcohol than people with a good SRH and good SH. I would have expected the opposite. Could this be discussed?

Response: Minor variations in alcohol consumption across the outcome groups are consistent with other Canadian studies that show level of drinking is slightly higher in those with better SRH (reference #1). The direction of the relationship between is yet to be established. Since alcohol was not an important factor and was only identified in one important node, for brevity, we did not discuss this finding. 

14. Table 1: Minor point: Sometimes it is not clear in which line certain numbers belong. One example: in the row “Perceived social class”, do the numbers 224 (2) 399 (7) belong to Lower class or Missing.

Response: Thank you. We edited all tables for a better formatting. 

15. In the conclusion of the paper nothing is mentioned about gender (or sex). I found that to be incoherent with both the title of the paper “does gender matter?” and the conclusion part of the abstract “Combining SRH and SH eliminated sex as a predictor of SRH/SH,…”

Response: Conclusion is rewritten - not sure how we missed addressing sex/gender in the conclusion and thank you for noticing this!

6. PLOS authors have the option to publish the peer review history of their article. If published, this will include your full peer review and any attached files.

Do you want your identity to be public for this peer review? For information about this choice, including consent withdrawal, please see our Privacy Policy.

Reviewer #1: No

Reviewer #2: No

Reviewer #3: Yes: Lisa Dandolo

---

## [Decision Letter · Decision Letter 1]

13 Sep 2023

PONE-D-23-10826R1Descriptive regression tree analysis of intersecting predictors of adult self-rated health: does gender matter? A cross-sectional study of Canadian adultsPLOS ONE

Dear Dr. Vafaei,

Thank you for submitting your manuscript to PLOS ONE. After careful consideration, we feel that it has merit but does not fully meet PLOS ONE’s publication criteria as it currently stands. Therefore, we invite you to submit a revised version of the manuscript that addresses the points raised during the review process.

The reviewers have appreciated your efforts to improve your work by implementing their comments. However, before you can recommend publication of your paper, you should take into account the minor comments made by the 3# reviewer in his latest review. Keep in mind that the reviewers have done a really thorough job, and their suggestions have contributed to the improvement of the final paper. 

We look forward to receiving your revised manuscript.

Kind regards,

Jordi Gumà, Ph.D.

Academic Editor

PLOS ONE

Journal Requirements:

Reviewers' comments:

Reviewer's Responses to Questions

**Comments to the Author**

1. If the authors have adequately addressed your comments raised in a previous round of review and you feel that this manuscript is now acceptable for publication, you may indicate that here to bypass the “Comments to the Author” section, enter your conflict of interest statement in the “Confidential to Editor” section, and submit your "Accept" recommendation.

Reviewer #3: (No Response)

2. Is the manuscript technically sound, and do the data support the conclusions?

Reviewer #3: Yes

3. Has the statistical analysis been performed appropriately and rigorously? 

Reviewer #3: Yes

4. Have the authors made all data underlying the findings in their manuscript fully available?

Reviewer #3: Yes

5. Is the manuscript presented in an intelligible fashion and written in standard English?

Reviewer #3: Yes

6. Review Comments to the Author

Reviewer #3: Thanks for revising the manuscript and considering and answering my previous comments.

I have four remaining points for a minor revision. After these minor points are addressed I would recommend to accept the paper.

1. Thanks for clarifying within the discussion, that the bivariate analyses of sex and SRH, and sex and SH were performed and both showed significant effects. I feel like these are decisive results in order to draw the conclusion of your paper. I would therefore really appreciate if you could actually report this data in the results section. It should not use up to much space and is important for the overall conclusion of the paper.

2. Would it be possible to show the inital distribution (before dichotomizing) of both the SRH and satisfaction with health outcome variables in a supplemental material document. This would surely benefit all readers that are interested in using similar outcome variables and would support your dichotomization decisions.

3. Thanks for clarifying in your response to my question which decision tree algorithm you used. I would recommend to include this one sentence: "we applied the Gini impurity with a minimal improvement set at 0.0001 and equal cost (the full SPSS syntax is available upon request)" into your methods section of the manuscript. It is only one sentence but can really help any readers that would like to use your technique for their analyses. Readers not interested in the techniqual details can always skipp reading the detailed methods section.

4. I still feel that the sentence "The regression tree correctly classified almost 90% of participants with positive outcomes

and resulted in 24 terminal nodes, ten of which were deemed important.”, somehow reports the data one sided, as only the better classification rate (for "positive outcomes") is reported and not the classification rate for participants with poor SRH/SH. You clarified this for me in your response to my question but did not include it in the manuscript.

7. PLOS authors have the option to publish the peer review history of their article (what does this mean?). If published, this will include your full peer review and any attached files.

Reviewer #3: **Yes: **Lisa Dandolo

---

## [Author Response · Author response to Decision Letter 1]

15 Oct 2023

6. Review Comments to the Author

Reviewer #3: Thanks for revising the manuscript and considering and answering my previous comments.

I have four remaining points for a minor revision. After these minor points are addressed I would recommend to accept the paper.

1. Thanks for clarifying within the discussion, that the bivariate analyses of sex and SRH, and sex and SH were performed and both showed significant effects. I feel like these are decisive results in order to draw the conclusion of your paper. I would therefore really appreciate if you could actually report this data in the results section. It should not use up to much space and is important for the overall conclusion of the paper.

Response: We added this sentence to the results section before explaining table 1 which include the composite SRH and SH variable “Before combining SRH and SH variables to generate a composite measure for our outcome, we examined their bivariate relationships with sex. Both showed significant associations according to the results of Chi-square test (p. value for SRH <0.001 and for SH=0.003).”

2. Would it be possible to show the initial distribution (before dichotomizing) of both the SRH and satisfaction with health outcome variables in a supplemental material document. This would surely benefit all readers that are interested in using similar outcome variables and would support your dichotomization decisions.

Response: thank you, a valid point since there is no established cut-off point for dichotomizing (See page 7 of the manuscript). We added these two histograms representing frequency distributions of original scales of SRH and SH as a supplement material to the revised submission. 

 3. Thanks for clarifying in your response to my question which decision tree algorithm you used. I would recommend to include this one sentence: "we applied the Gini impurity with a minimal improvement set at 0.0001 and equal cost (the full SPSS syntax is available upon request)" into your methods section of the manuscript. It is only one sentence but can really help any readers that would like to use your technique for their analyses. Readers not interested in the techniqual details can always skipp reading the detailed methods section.

Response: Agree, this one sentence contains important information about our analytic method. Added to the ‘statistical analysis’ subsection of methods (page 10). 

4. I still feel that the sentence "The regression tree correctly classified almost 90% of participants with positive outcomes and resulted in 24 terminal nodes, ten of which were deemed important.”, somehow reports the data one sided, as only the better classification rate (for "positive outcomes") is reported and not the classification rate for participants with poor SRH/SH. You clarified this for me in your response to my question but did not include it in the manuscript.

Response: we reported the 76% overall correct classification in the methods. We agree that to avoid a one-sided report of data a more detailed description of misclassification(s) is warranted. We edited the above statement in page 12 to: “The regression tree correctly classified 93.3% of participants with positive outcomes and resulted in 24 terminal nodes, ten of which were deemed important. Correct classification for those with negative outcome was 35% yielded a total correct classification of 76%.”

---

## [Editor Report · Decision Letter 2]

24 Oct 2023

Descriptive regression tree analysis of intersecting predictors of adult self-rated health: does gender matter? A cross-sectional study of Canadian adults

PONE-D-23-10826R2

Dear Dr. Vafaei,

We’re pleased to inform you that your manuscript has been judged scientifically suitable for publication and will be formally accepted for publication once it meets all outstanding technical requirements.

Kind regards,

Jordi Gumà, Ph.D.

Academic Editor

PLOS ONE
---

## [Editor Report · Acceptance letter]

6 Nov 2023

PONE-D-23-10826R2 

Descriptive regression tree analysis of intersecting predictors of adult self-rated health: does gender matter? A cross-sectional study of Canadian adults 

Dear Dr. Vafaei:

I'm pleased to inform you that your manuscript has been deemed suitable for publication in PLOS ONE. Congratulations! Your manuscript is now with our production department. 

Kind regards, 

on behalf of

Dr. Jordi Gumà 

Academic Editor

PLOS ONE